analysis

parameter identification, Hilbert transform, moment of velocity

**Author for correspondence:**
M. Dorraki
e-mail: mohsen.dorraki@adelaide.edu.au

# Parameter identification using moment of velocity

M. Dorraki[1,2], M. S. Islam[1], A. Allison[1,2] and D. Abbott[1,2]

[1]School of Electrical and Electronic Engineering, and [2]Centre for Biomedical Electrical Engineering (CBME), The University of Adelaide, Adelaide, South Australia 5005, Australia

MD, 0000-0002-6675-3393; MSI, 0000-0003-0578-4732;
AA, 0000-0003-3865-511X; DA, 0000-0002-0945-2674

Many physical systems can be adequately modelled using a second-order approximation. Thus, the problem of system identification often reduces to the problem of estimating the position of a single pair of complex–conjugate poles. This paper presents a convenient but approximate technique for the estimation of the position of a single pair of complex–conjugate poles, using the moment of velocity (MoV). The MoV is a Hilbert transform based signal processing tool that addresses the shortcomings of instantaneous frequency. We demonstrate that the MoV can be employed for parameter identification of a dynamical system. We estimate the damping coefficient and oscillation frequency via MoV of the impulse response.

## 1. Introduction

In many real-world problems, lack of a dynamic representation of the system under analysis is a major difficulty. Therefore, system identification can be considered as a helpful approach to provide a dynamical model for the system. In order to obtain the model, one may stimulate the system with a specific signal and use the output for deriving a state space model in the time domain or alternatively a transfer function model in the frequency domain. In previous studies, some specific signals such as impulse [1], step [2,3] and ramp [4] inputs are employed to explore the response of dynamical systems to external stimuli.

Parameter identification is a longstanding theme of system identification [5,6]. The field of parameter identification has emerged as a significant area of engineering, and there exists a large number of studies. Some useful tools such as integral equations [7], neural networks [8], least-squares approaches [9], Newton iteration [10] and genetic algorithms [11] have been reported to assist with the parameter identification problem. The application of the Hilbert transform for parameter identification has been widely investigated over the last few decades. A number of these previous studies are briefly summarized in the following.

In general, the methods take advantage of instantaneous frequency (IF) and instantaneous amplitude (IA) of the impulse response to identify the damping coefficient and oscillation frequency separately, and in these methods the IF and IA are obtained from the Hilbert transform. In one of the early studies, two points on impulse response curve whose IA is reduced by −8.7 dB are used to estimate damping coefficient and oscillation frequency [12]. In another study, both instantaneous phase and amplitude is employed to estimate the parameters [13]. To reduce the influence of noise, the previous method is modified by employing a least-squares regression on the IA for a truncated signal [14].

As an application in mechanical engineering, a time domain non-parametric method for nonlinear vibration system identification based on the Hilbert transform has been introduced [15]. Using this approach, both the oscillation frequency and also the real nonlinear elastic force characteristics can be extracted. Based on the Hilbert–Huang spectral analysis, a method was proposed to identify multi-degree-of-freedom linear systems using the measured free vibration signal [16]. The approach uses a linear least-square fit algorithm to identify the oscillation frequency and damping coefficient from the IA and phase.

A parameter identification approach based on empirical mode decomposition, the random decrement technique, and the Hilbert–Huang transform was also proposed [17]. The study used ambient wind vibration data in order to estimate the damping coefficient and oscillation frequency in tall buildings. Moreover, an approach for parameter identification in nonlinear vibrating systems has been presented based on measured signals for free and forced vibration regimes [18]. That experimental method considered the application of the Hilbert transform for identification of nonlinearities in stiffness and damping characteristics of a mechanical vibrating system.

In this paper, we propose a new moment of velocity (MoV) approach to address the problem of parameter identification. In particular, we show that the MoV is a convenient but approximate approach for the estimation of the position of a complex–conjugate pair of poles on the $s$ plane. Moment of velocity as a signal processing tool, is considered a reliable alternative to IF as it suppresses large spikes that often clutter the IF signal [19]. In addition to IF, IA is incorporated in MoV. For illustrative purposes, we apply MoV to the impulse response of a dynamical system that behaves as a pure exponential. The signal can be unwrapped using the $\ln(\cdot)$ function, and following a linear least-square fit procedure, the damped parameters may be obtained simultaneously from the slope and $y$-intercept of the fitted line. Moreover, in order to investigate the impact of noise on the estimated parameters, a noise experiment is conducted for various levels of SNR. The results show that errors in the estimated parameters are tolerable for SNRs of 30 dB or better.

## 2. The moment of velocity

In mathematics and signal processing, the Hilbert transform is a specific linear operator that obtains a real variable function $y(t)$ and returns another real variable function $H[y(t)]$. The mathematical definition of the Hilbert transform is written usually in the form [20–22]:

$$H[y(t)] = \frac{1}{\pi} \int_{-\infty}^{+\infty} \frac{y(\tau)}{t - \tau} \, d\tau. \tag{2.1}$$

The Hilbert transform also can be defined as a convolution between the signal and $1/\pi t$:

$$H[y(t)] = y(t) \otimes \frac{1}{\pi t}. \tag{2.2}$$

The IA is defined as the magnitude or absolute value of the analytic function, i.e. $z(t) = y(t) + iH[y(t)]$. Therefore, the IA is given by

$$e(t) = \sqrt{y(t)^2 + H[y(t)]^2}. \tag{2.3}$$

The IF of a signal is a function of time and also a measure of the frequency corresponding to a particular time component of the signal. For a real signal, $y(t)$, the IF, $f(t)$, is defined as

$$f(t) = \frac{1}{2\pi} \frac{d\phi(t)}{dt}, \tag{2.4}$$

where $\phi(t)$ is the instantaneous phase of analytic signal $z(t)$. Therefore, the IF may be written in following form:

$$
\begin{aligned}
f(t) &= \frac{1}{2\pi}\frac{\mathrm{d}}{\mathrm{d}t}\left[\arctan\left(\frac{\mathrm{H}[y(t)]}{y(t)}\right)\right] \\
&= \frac{y(t)(\mathrm{d}\mathrm{H}[y(t)]/\mathrm{d}t) - \mathrm{H}[y(t)](\mathrm{d}y(t)/\mathrm{d}t)}{y(t)^2 + \mathrm{H}[y(t)]^2}.
\end{aligned}
\tag{2.5}
$$

Although IF is widely used in signal processing, in many cases using IF as a signal processing tool is challenging to interpret. Also, IF is sensitive to noise because in the formula for calculating IF the numerator is divided by a dominator that can be very small, and consequently IF as a function of time can become cluttered with spikes.

In order to overcome the deficiencies, the MoV is introduced [19]. The MoV is very similar to IF except that the denominator of the instantaneous frequency equation is removed. Therefore, this significantly avoids singularities in the phase space and clutter in the $f(t)$ waveform. The MoV is defined as [19]

$$
\text{moment of velocity} = y(t)\frac{\mathrm{d}\mathrm{H}[y(t)]}{\mathrm{d}t} - \mathrm{H}[y(t)]\frac{\mathrm{d}y(t)}{\mathrm{d}t}.
\tag{2.6}
$$

# 3. The impulse response of a second-order system

In control theory, physical systems are generally described mathematically in terms of linear systems of ordinary differential equations [23]. When these equations are transformed using integral transforms, such as the Laplace or Fourier, then physical systems are modelled using finite rational polynomials in an auxiliary variable, $s = j\omega$. Therefore, the transfer function is a rational function in the complex variable, that is

$$
\text{Transfer function} = \frac{\text{output}(s)}{\text{input}(s)} = \frac{P(s)}{Q(s)}.
\tag{3.1}
$$

The zeros of the polynomial, $Q(s)$, are called poles and correspond to responses that have finite output for zero input. It is widespread for one mode to dominate the response of the whole system. It is also common for this mode to be of a damped oscillatory type, corresponding to a single pair of complex–conjugate poles. This can occur whenever the potential energy function of the system possesses a local minimum [24]. In this case, one can approximate a large complicated system, with many poles and zeros, by a simple second-order system with a single pair of complex–conjugate poles—this is what is referred to as a second-order approximation. Various mechanical or electrical systems may be realistically modelled using a second-order approximation. Thus,

$$
\text{Transfer function} \approx \frac{a_2 s^2 + a_1 s + a_0}{s^2 + 2\alpha s + \omega_0^2}.
\tag{3.2}
$$

In order to model the behaviour of a real physical system, using an approximate second-order model, it is necessary to estimate the position of the pole pair. This can potentially be carried out in the frequency domain, by stimulating the system with a sinusoidal source and then measuring the magnitude and phase of the response at different frequencies; however, this is often not practical. There are situations when the only practical sources are step functions, $u(t)$, or impulses $\delta(t)$. Therefore, we may stimulate the system with steps or impulses and then sample the response in the time domain. The impulse response of a second-order system generally is written as

$$
y(t) = A\,\mathrm{e}^{-\alpha t}\cos(\omega_d t) + B\,\mathrm{e}^{-\alpha t}\sin(\omega_d t),
\tag{3.3}
$$

where $\omega_d^2 = \omega_0^2 - \alpha^2$.

The problem of plant parameter identification then becomes equivalent to asking: How does one estimate the position of the pair of complex–conjugate poles if the only data at our disposal is a set of time-domain samples of the response of the system to steps or impulses? As an illustration, consider a bell struck with a hammer and the sound is recorded as it gradually decays. The estimation of the damped frequency of oscillation $\omega_d$ and the damping coefficient $\alpha$ using only the data from the sound recording is desirable.

Considering the distribution of the errors of measurement, the maximum likelihood method can potentially be applied to estimate the parameters $\omega_d$ and $\alpha$. If the errors are known to be the result of

a large number of uncorrelated random effects, then one can potentially apply the Central Limit Theorem and assume the errors to have a Gaussian distribution. Thus, the problem of plant identification reduces to a nonlinear least-squares estimation problem [25]. The difficulty with this approach is that the resulting equations are nonlinear and need to be solved iteratively, using a numerical method such as gradient descent. A further weakness of this approach is that it is an exact solution to an approximation of the real problem. Therefore, for certain types of problems, it is reasonable to have a ready but approximate solution to the approximate, second-order, problem—the MoV method provides this, as described in the following.

# 4. Parameter identification using MoV

The impulse response described in equation (3.3) is equivalent to

$$y(t) = C e^{-\alpha t} \cos(\omega_d t - \Phi), \tag{4.1}$$

where $C = \sqrt{A^2 + B^2}$, $\cos(\Phi) = A/\sqrt{A^2 + B^2}$ and $\sin(\Phi) = B/\sqrt{A^2 + B^2}$. The constant $C$ can be determined using initial conditions when $t$ equals zero in the impulse response. This type of function will apply whenever the input to the system is zero. If the input is a finite sum of step and impulse functions, then the input will be zero for most of the time. There will be abrupt changes in $C$ and $\Phi$ but the parameters, $\alpha$ and $\omega_d$ will be constant as long as the structure of the plant is maintained.

Bedrosian's theorem states that the Hilbert transform of the product of a low-pass and a high-pass signal with non-overlapping spectra is given by the product of the low-pass signal and the Hilbert transform of the high-pass signal [21]. Appling Bedrosian's theorem and the shifting property to equation (4.1) implies that the Hilbert transform of $y(t)$ is

$$H[y(t)] = C e^{-\alpha t} \sin(\omega_d t - \Phi). \tag{4.2}$$

Considering equation (2.6), the MoV of $y(t)$ is defined as

$$\begin{aligned}
\mathrm{MoV}[y(t)] &= \frac{1}{2\pi}\frac{\mathrm{d}}{\mathrm{d}t}\left[\arctan\left(\frac{H[y(t)]}{y(t)}\right)\right] \cdot (y(t)^2 + H[y(t)]^2), \\
&= \frac{1}{2\pi}C^2 e^{-2\alpha t} \cdot \frac{\mathrm{d}}{\mathrm{d}t}(\omega_d t - \Phi), \\
&= \frac{1}{2\pi}C^2 \omega_d e^{-2\alpha t}.
\end{aligned} \tag{4.3}$$

This signal is a pure exponential function and can essentially be unwrapped using the $\ln(\cdot)$ function. Thus,

$$\ln(\mathrm{MoV}[y(t)]) = -2\alpha t + \ln\left(\frac{C^2}{2\pi}\omega_d\right). \tag{4.4}$$

Therefore, the damped oscillation frequency $\omega_d$ can be obtained from the y-intercept and the damping coefficient $\alpha$ may be obtained from the slope of equation (4.4). To achieve this, a linear least-square fit procedure is used to estimate the slope and y-intercept. The coefficients of the fitted line with equation of $p(x) = p_1 x + p_0$ is used for estimating of the parameters:

$$\alpha = -\frac{p_1}{2} \tag{4.5}$$

and

$$\omega_d = \frac{2\pi e^{p_0}}{C^2}. \tag{4.6}$$

This allows us to directly estimate the parameters $\alpha$ and $\omega_d$. It may be seen that the use of the difference operation has removed all reference to $\Phi$.

# 5. Numerical results and noise analysis

To examine our approach, a second-order system using known damped parameters is simulated. In addition, we assess the performance of proposed parameter approach identification on two real-world problems.

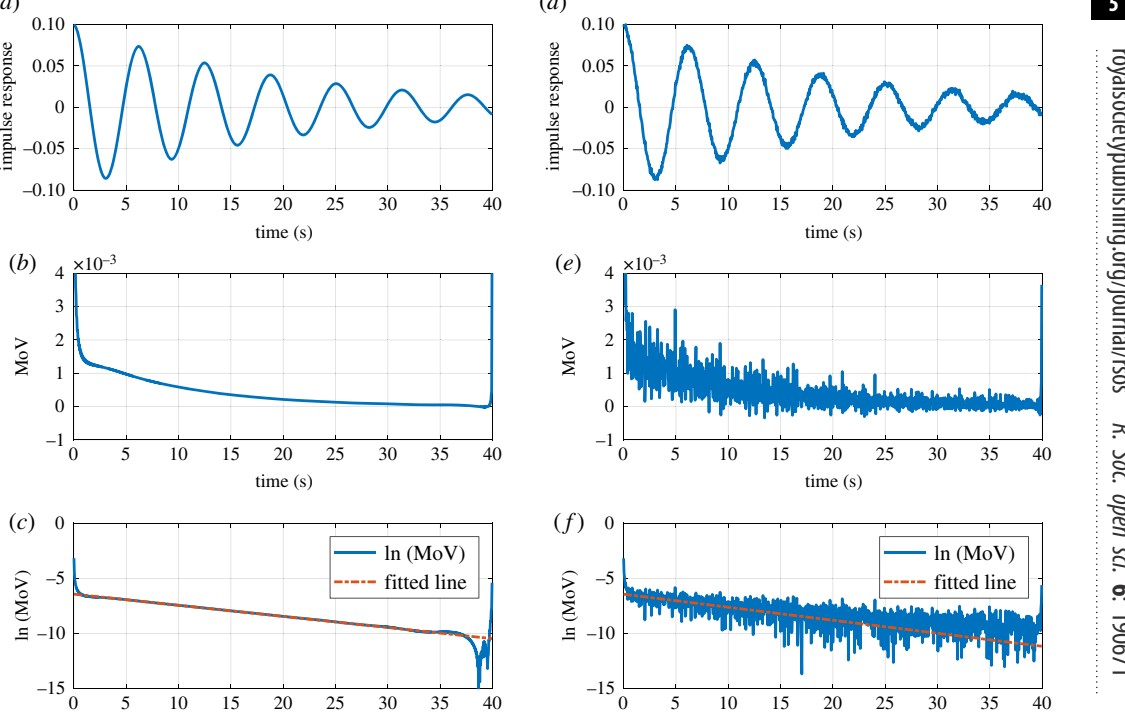

**Figure 1.** Parameter identification using MoV. (*a*) The impulse response of a second-order system with damping coefficient $\alpha = 0.05$ and oscillation frequency of $\omega_d = 1$. (*b*) The MoV of the impulse response. (*c*) Taking logarithm of MoV of impulse response and fitting a line on output signal. (*d*) Adding 25 dB SNR noise to the impulse response. (*e*) The moment of velocity of the noisy impulse response. (*f*) Taking logarithm of MoV of noisy impulse response and fitting a line on the output signal.

## 5.1. Simulated system

First, a second-order system is simulated. The quality factor of the system is $Q = 10$ and so the theoretical value of the damping coefficient is $\alpha = 0.05$. Moreover, the characteristic frequency of the simulated plant is $\omega_d = 1.0$. After applying the MoV approach, the estimated damping coefficient is obtained $\alpha_{est} = 0.0504$ and the estimated oscillation frequency is obtained $\omega_{est} = 1.0009$.

The system responses usually are contaminated with noise introduced by the channel and sampling deficiencies. The noise may exert a considerable impact on the outcome of the parameter identification approach. A comprehensive noise analysis is presented in [19] for MoV and other Hilbert-based tools such as IF that explains the relative robustness of MoV in noisy conditions. To assess the performance of the MoV parameter identification approach in noisy conditions, we intentionally add 25 dB SNR noise to the impulse response. The estimated damping coefficient in noisy conditions is obtained $\alpha_{est} = 0.0580$ and the estimated oscillation frequency is obtained $\omega_{est} = 1.0090$. The procedure is shown in figure 1. Our simulations suggest that MoV is a convenient technique for the estimation of the position of a single pair of complex–conjugate poles even under noisy conditions. In order to evaluate the approach under noisy conditions, the percentage of errors in $\alpha_{est}$ and also $\omega_{est}$ versus SNR is demonstrated in figure 2. As expected, it can be seen that an increase in SNR results in reduced error.

## 5.2. Real-world problems

To assess the proposed parameter identification method for real-world problems, two time series related to a free vibration test and an industrial production response after the 1990 earthquake are employed.

A free vibration test of a submerged pipeline was established by giving an initial displacement in the middle of the pipeline, and the response was recorded over time [26]. The damping coefficient $\alpha = 0.4824$ and the damped oscillation frequency $\omega_d = 20.1062$ are determined from the experiment. The time series is shown in figure 3*a*, and the corresponding MoV and fitted line are also illustrated in figure 3*b,c*. The estimated damping coefficient for this time series is obtained as $\alpha_{est} = 0.4725$ and the estimated oscillation frequency is obtained $\omega_{est} = 20.3184$.

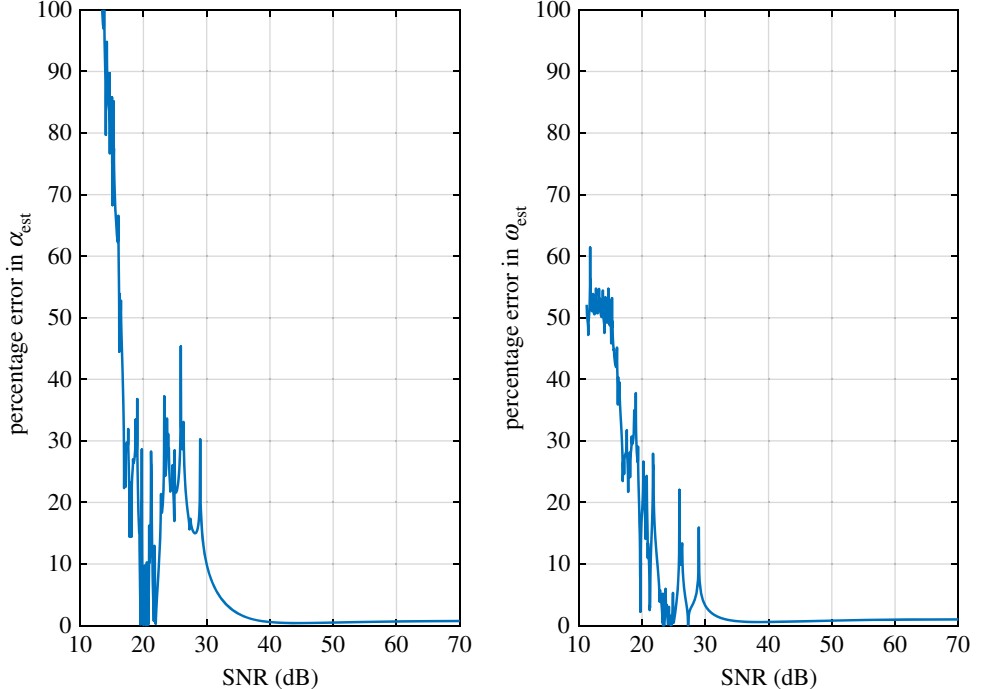

**Figure 2.** Absolute values of percentage error in $\alpha_{est}$ and $\omega_{est}$ versus SNR.

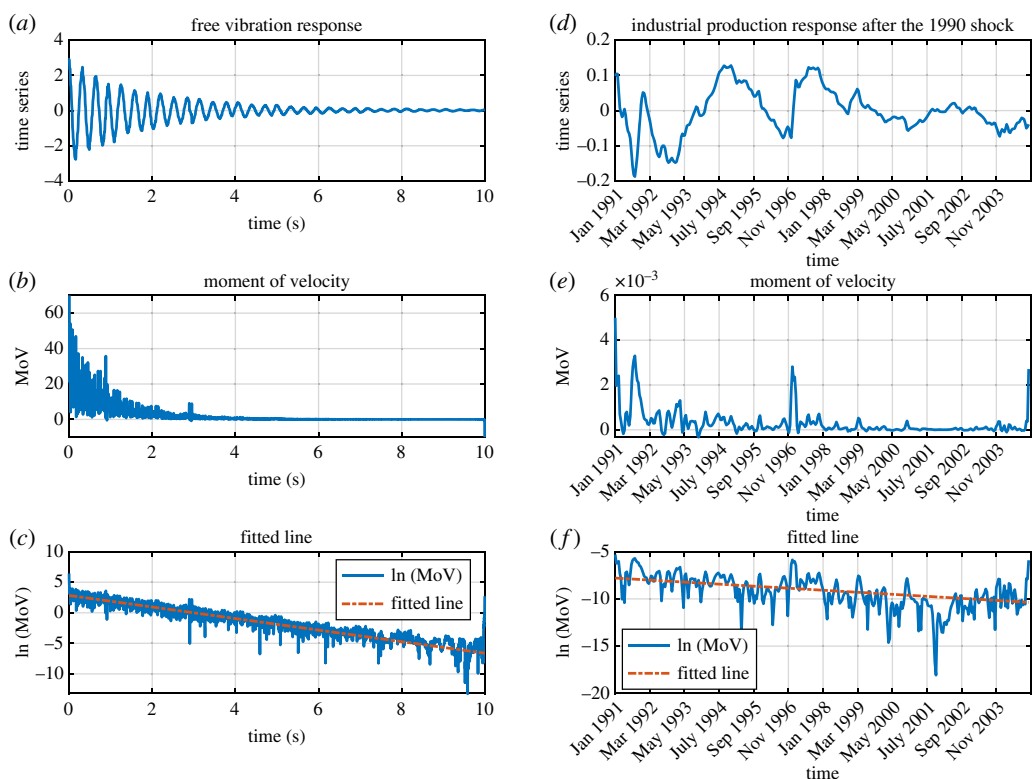

**Figure 3.** Assessing the MoV approach for real-world problems. (*a*) The response corresponding to a free vibration test with damping coefficient $\alpha = 0.4824$ and oscillation frequency of $\omega_d = 20.1062$. (*b*) The moment of velocity of the free vibration response. (*c*) Taking logarithm of MoV and fitting a line on output signal. (*d*) An industrial production time series with damping coefficient $\alpha = 0.0077$ and oscillation frequency of $\omega_d = 0.1387$. (*e*) The MoV of the industrial production time series. (*f*) Taking logarithm of MoV and fitting a line on the output signal.

In the same way, another dynamic response related to an industrial production response after the 1990 earthquake is used for assessing the MoV parameter identification approach. The oscillatory behaviour of industrial production is shown in figure 3*d*. The damping coefficient $\alpha = 0.0077$ and the

**Table 1.** The experimental and estimated damping parameters using our proposed MoV approach and a traditional method [14] for free vibration test [26] and industrial production [27] problems.

| parameters | free vibration | | | industrial production | | |
| --- | --- | --- | --- | --- | --- | --- |
| | experimental [26] | MoV | traditional [14] | experimental [27] | MoV | traditional [14] |
| $\alpha$ | 0.4824 | 0.4725 | 0.4618 | 0.0077 | 0.0061 | 0.0057 |
| | | err = 2.05% | err = 2.31% | | err = 18.0328% | err = 26.3158% |
| $\omega_d$ | 20.1062 | 20.3184 | 19.8520 | 0.1387 | 0.1770 | 0.1038 |
| | | err = 1.04% | err = 1.28% | | err = 21.6384% | err = 33.6224% |

damped oscillation frequency $\omega_d = 0.1387$ are obtained [27]. Applying the MoV method to the time series provides the estimated damping coefficient $\alpha_{est} = 0.0061$ and the estimated oscillation frequency $\omega_{est} = 0.1770$. The MoV and the corresponding fitted line are shown in figure 3*e,f* .

We also evaluate both MoV and the approach of Agneni *et al.* [14] as a well-known traditional parameter identification method and modified version of [12,13]. The dynamic properties of the mechanical and economic systems are listed in table 1, which were obtained through experimental testing and parameter estimation. The performance of MoV versus the method of Agneni *et al.* is evaluated in the table 1 that shows MoV is more reliable than Agneni's approach particularly for the industrial production case.

# 6. Conclusion

A formulation based on the MoV is presented for convenient system parameter identification. Our approach has potential application in continuous real time approximation such as in monitoring and tracking control. The simulations suggest that MoV is a convenient method for estimating the position of a single pair complex–conjugate poles. Errors in the estimated parameters are tolerable for SNRs of 30 dB or better. A limitation of the method is that errors become large in the 0 dB to 30 dB SNR range. It may be of interest for future studies to investigate this form of parameter identification a high noise levels, after first adopting a signal denoising algorithm.

Data accessibility. We have made the MATLAB code for parameter identification openly available on Github at https://github.com/Dorraki/Parameter-Identification-Using-MoV.
Authors' contributions. M.D., D.A. and A.A. developed the main idea; M.D. performed the analyses, and wrote the paper; D.A., A.A., M.S.I. and M.D. conceived the study; D.A. and A.A. supervised the study; all authors proofed the manuscript and contributed to interpretation of results.
Competing interests. Prof. Derek Abbott is a Board Member of Royal Society Open Science.
Funding. No funding has been received for this article.

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
