## [Reviewer comments · Royal Society Open Science]

Review History

RSOS-190671.R0 (Original submission)

Review form: Reviewer 1

Is the manuscript scientifically sound in its present form?

No

Are the interpretations and conclusions justified by the results?

No

Is the language acceptable?

No

Is it clear how to access all supporting data?

Not Applicable

Do you have any ethical concerns with this paper?

No

Have you any concerns about statistical analyses in this paper?

No

Recommendation?

Reject

Comments to the Author(s)

Parameter identification using moment of velocity

The authors develop a new parameter identification method for system identification. The goal of the authors is laudable. However, I am not convinced by the example shown. The investigators should show how this works on real data. They need to provide concrete examples. They also need to use real noise. The simulation does not show how the system will work on actual data.

There are also some grammatical errors which need remedy for example -
even under the noisy conditions
even under noisy conditions

Matlab should be MATLAB

'A limitation of the method is that errors become large in the 0 dB to 30 dB SNR range.'
Authors need to show what happens and why and how to avoid it.

Review form: Reviewer 2

Is the manuscript scientifically sound in its present form?

No

Are the interpretations and conclusions justified by the results?

Yes

Is the language acceptable?

Yes

Is it clear how to access all supporting data?

Yes

Do you have any ethical concerns with this paper?

No

Have you any concerns about statistical analyses in this paper?

No

Recommendation?

Accept with minor revision (please list in comments)

Comments to the Author(s)

1. The paper discusses the parameter identification problems. In fact, there exist many identification methods in the literature. Typical methods include the iterative identification methods [i1-i5] and the recursive identification methods [r1-r5]. These important works should be

surveyed and mentioned in the paper, they play a very important role in system identification and parameter estimation.

2. Specifically, some classic transfer function parameter identification methods are very related to the authors' current submission, they are very useful for system analysis and control [t1-t5].

3. The paper is organized well and the paper contains some materials worth of publication.

4. Some references are too old and may be removed. The references should use the RSOS style.

5 The paper can be accepted after revision according to the above comments.

[t1] Application of the Newton iteration algorithm to the parameter estimation for dynamical systems, *Journal of Computational and Applied Mathematics* 288 (2015) 33-43.

[t2] Parameter estimation and controller design for dynamic systems from the step responses based on the Newton iteration, *Nonlinear Dynamics* 79 (3) (2015) 2155-2163.

[t3] The damping iterative parameter identification method for dynamical systems based on the sine signal measurement, *Signal Processing* 120 (2016) 660-667.

[t4] Hierarchical parameter estimation for the frequency response based on the dynamical window data, *International Journal of Control Automation and Systems* 16 (4) (2018) 1756-1764.

[t5] A proportional differential control method for a time-delay system using the Taylor expansion approximation, *Applied Mathematics and Computation* 236 (2014) 391-399.

[i1] Gradient based and least-squares based iterative identification methods for OE and OEMA systems, *Digital Signal Processing* 20 (3) (2010) 664-677.

[i2] Gradient-based and least-squares-based iterative algorithms for Hammerstein systems using the hierarchical identification principle, *IET Control Theory and Applications* 7 (2) (2013) 176-184.

[i3] Two-stage least squares based iterative estimation algorithm for CARARMA system modeling, *Applied Mathematical Modelling* 37 (7) (2013) 4798-4808.

[i4] Decomposition based fast least squares algorithm for output error systems, *Signal Processing* 93 (5) (2013) 1235-1242.

[i5] Gradient based and least squares based iterative estimation algorithms for multi-input multi-output systems, *Proceedings of the Institution of Mechanical Engineers, Part I: Journal of Systems and Control Engineering* 226 (1) (2012) 43-55.

[r1] Coupled-least-squares identification for multivariable systems, *IET Control Theory and Applications* 7 (1) (2013) 68-79.

[r2] Hierarchical multi-innovation stochastic gradient algorithm for Hammerstein nonlinear system modeling, *Applied Mathematical Modelling* 37 (4) (2013) 1694-1704.

[r3] States based iterative parameter estimation for a state space model with multi-state delays using decomposition, *Signal Processing* 106 (2015) 294-300.

[r4] State filtering and parameter estimation for linear systems with d-step state-delay, *IET Signal Processing* 8 (6) (2014) 639-646.

[r5] Combined state and least squares parameter estimation algorithms for dynamic systems, *Applied Mathematical Modelling* 38 (1) (2014) 403-412.

Review form: Reviewer 3 (Jason Ralph)

Is the manuscript scientifically sound in its present form?

Yes

Are the interpretations and conclusions justified by the results?

Yes

Is the language acceptable?

Yes

Is it clear how to access all supporting data?

Not Applicable

Do you have any ethical concerns with this paper?

No

Have you any concerns about statistical analyses in this paper?

No

Recommendation?

Accept with minor revision (please list in comments)

Comments to the Author(s)

The manuscript introduces an interesting method for the characterisation of a second order impulse response, using the Hilbert transform and the Moment of Velocity. The method appears to be novel, it is certainly new to me. The description of the techniques employed is clear and concise, and easy to follow. The results are good and the method certainly shows promise. The only thing that I suggest that the authors look to include is a direct comparison of their technique against a standard text book method. In addition, I think that the paper would benefit from a comment on the robustness of the method if is applied to situations where the the impulse response is not strictly second order. Other than these two points, I would recommend publication.

Decision letter (RSOS-190671.R0)

07-Aug-2019

Dear Mr Dorraki,

The editors assigned to your paper ("Parameter identification using moment of velocity (MoV)") have now received comments from reviewers. We would like you to revise your paper in accordance with the referee and Associate Editor suggestions which can be found below (not including confidential reports to the Editor). Please note this decision does not guarantee eventual acceptance.

Please submit a copy of your revised paper before 30-Aug-2019. Please note that the revision deadline will expire at 00.00am on this date. If we do not hear from you within this time then it will be assumed that the paper has been withdrawn. In exceptional circumstances, extensions may be possible if agreed with the Editorial Office in advance. We do not allow multiple rounds of revision so we urge you to make every effort to fully address all of the comments at this stage. If deemed necessary by the Editors, your manuscript will be sent back to one or more of the original reviewers for assessment. If the original reviewers are not available, we may invite new reviewers.

- Data accessibility

If you wish to submit your supporting data or code to Dryad (<http://datadryad.org/>), or modify your current submission to dryad, please use the following link:
<http://datadryad.org/submit?journalID=RSOS&manu=RSOS-190671>

- Competing interests

- Authors' contributions

- Acknowledgements

- Funding statement

Kind regards,
Lianne Parkhouse
Editorial Coordinator
Royal Society Open Science
openscience@royalsociety.org

on behalf of Professor R. Kerry Rowe (Subject Editor)
openscience@royalsociety.org

Associate Editor's comments:

Thanks for your patience with this paper. The journal struggled a little to find suitable reviewers to assess the piece. Given the slightly divergent opinions on the work, we'd like to offer you an opportunity of a revision to tackle the concerns raised by the reviewers. If you can provide reasonable responses and a revised manuscript in the revision, we will return the paper to the reviewers for further consideration. Good luck!

Reviewers' Comments to Author:

Reviewer: 1
Comments to the Author(s)

Parameter identification using moment of velocity

The authors develop a new parameter identification method for system identification. The goal of the authors is laudable. However, I am not convinced by the example shown. The investigators should show how this works on real data. They need to provide concrete examples. They also need to use real noise. The simulation does not show how the system will work on actual data.

There are also some grammatical errors which need remedy for example -
even under the noisy conditions
even under noisy conditions

Matlab should be MATLAB

'A limitation of the method is that errors become large in the 0 dB to 30 dB SNR range.'
Authors need to show what happens and why and how to avoid it.

Reviewer: 2

Comments to the Author(s)

1. The paper discusses the parameter identification problems. In fact, there exist many identification methods in the literature. Typical methods include the iterative identification methods [i1-i5] and the recursive identification methods [r1-r5]. These important works should be surveyed and mentioned in the paper, they play a very important role in system identification and parameter estimation.

2. Specifically, some classic transfer function parameter identification methods are very related to the authors' current submission, they are very useful for system analysis and control [t1-t5].

3. The paper is organized well and the paper contains some materials worth of publication.

4. Some references are too old and may be removed. The references should use the RSOS style.

5 The paper can be accepted after revision according to the above comments.

[t1] Application of the Newton iteration algorithm to the parameter estimation for dynamical systems, *Journal of Computational and Applied Mathematics* 288 (2015) 33-43.

[t2] Parameter estimation and controller design for dynamic systems from the step responses based on the Newton iteration, *Nonlinear Dynamics* 79 (3) (2015) 2155-2163.

[t3] The damping iterative parameter identification method for dynamical systems based on the sine signal measurement, *Signal Processing* 120 (2016) 660-667.

[t4] Hierarchical parameter estimation for the frequency response based on the dynamical window data, *International Journal of Control Automation and Systems* 16 (4) (2018) 1756-1764.

[t5] A proportional differential control method for a time-delay system using the Taylor expansion approximation, *Applied Mathematics and Computation* 236 (2014) 391-399.

[i1] Gradient based and least-squares based iterative identification methods for OE and OEMA systems, *Digital Signal Processing* 20 (3) (2010) 664-677.

[i2] Gradient-based and least-squares-based iterative algorithms for Hammerstein systems using the hierarchical identification principle, *IET Control Theory and Applications* 7 (2) (2013) 176-184.

[i3] Two-stage least squares based iterative estimation algorithm for CARARMA system modeling, *Applied Mathematical Modelling* 37 (7) (2013) 4798-4808.

[i4] Decomposition based fast least squares algorithm for output error systems, *Signal Processing* 93 (5) (2013) 1235-1242.

[i5] Gradient based and least squares based iterative estimation algorithms for multi-input multi-output systems, *Proceedings of the Institution of Mechanical Engineers, Part I: Journal of Systems and Control Engineering* 226 (1) (2012) 43-55.

[r1] Coupled-least-squares identification for multivariable systems, *IET Control Theory and Applications* 7 (1) (2013) 68-79.

[r2] Hierarchical multi-innovation stochastic gradient algorithm for Hammerstein nonlinear system modeling, *Applied Mathematical Modelling* 37 (4) (2013) 1694-1704.

[r3] States based iterative parameter estimation for a state space model with multi-state delays using decomposition, *Signal Processing* 106 (2015) 294-300.

[r4] State filtering and parameter estimation for linear systems with d-step state-delay, *IET Signal Processing* 8 (6) (2014) 639-646.

[r5] Combined state and least squares parameter estimation algorithms for dynamic systems, *Applied Mathematical Modelling* 38 (1) (2014) 403-412.

Reviewer: 3

Comments to the Author(s)

The manuscript introduces an interesting method for the characterisation of a second order impulse response, using the Hilbert transform and the Moment of Velocity. The method appears to be novel, it is certainly new to me. The description of the techniques employed is clear and

concise, and easy to follow. The results are good and the method certainly shows promise. The only thing that I suggest that the authors look to include is a direct comparison of their technique against a standard text book method. In addition, I think that the paper would benefit from a comment on the robustness of the method if is applied to situations where the the impulse response is not strictly second order. Other than these two points, I would recommend publication.

Author's Response to Decision Letter for (RSOS-190671.R0)

See Appendix A.

RSOS-190671.R1 (Revision)

Review form: Reviewer 1

Is the manuscript scientifically sound in its present form?

Yes

Are the interpretations and conclusions justified by the results?

Yes

Is the language acceptable?

Yes

Do you have any ethical concerns with this paper?

No

Have you any concerns about statistical analyses in this paper?

No

Recommendation?

Accept as is

Comments to the Author(s)

The authors satisfactorily implemented my suggestions.

Review form: Reviewer 2

Is the manuscript scientifically sound in its present form?

No

Are the interpretations and conclusions justified by the results?

Yes

Is the language acceptable?

Yes

Do you have any ethical concerns with this paper?

No

Have you any concerns about statistical analyses in this paper?

No

Recommendation?

Accept with minor revision (please list in comments)

Comments to the Author(s)

The revision has some improvement. The paper is acceptable.

But the authors miss several important references in the first round review.

This should be added before the paper is published.

Decision letter (RSOS-190671.R1)

03-Sep-2019

Dear Mr Dorraki:

On behalf of the Editors, I am pleased to inform you that your Manuscript RSOS-190671.R1 entitled "Parameter identification using moment of velocity (MoV)" has been accepted for publication in Royal Society Open Science subject to minor revision in accordance with the referee suggestions. Please find the referees' comments at the end of this email.

The reviewers and Subject Editor have recommended publication, but also suggest some minor revisions to your manuscript. Therefore, I invite you to respond to the comments and revise your manuscript.

- **Ethics statement**

- **Data accessibility**

It is a condition of publication that all supporting data are made available either as supplementary information or preferably in a suitable permanent repository. The data accessibility section should state where the article's supporting data can be accessed. This section should also include details, where possible of where to access other relevant research materials such as statistical tools, protocols, software etc can be accessed. If the data has been deposited in an external repository this section should list the database, accession number and link to the DOI for all data from the article that has been made publicly available. Data sets that have been

deposited in an external repository and have a DOI should also be appropriately cited in the manuscript and included in the reference list.

<http://datadryad.org/submit?journalID=RSOS&manu=RSOS-190671.R1>

- **Competing interests**

- **Authors' contributions**

- **Acknowledgements**

- **Funding statement**

Because the schedule for publication is very tight, it is a condition of publication that you submit the revised version of your manuscript before 12-Sep-2019. Please note that the revision deadline will expire at 00.00am on this date. If you do not think you will be able to meet this date please let me know immediately.

When submitting your revised manuscript, you will be able to respond to the comments made by the referees and upload a file "Response to Referees" in "Section 6 - File Upload". You can use this

to document any changes you make to the original manuscript. In order to expedite the processing of the revised manuscript, please be as specific as possible in your response to the referees.

on behalf of Prof R. Kerry Rowe (Subject Editor)
openscience@royalsociety.org

Reviewer comments to Author:
Reviewer: 2

Comments to the Author(s)
The revision has some improvement. The paper is acceptable.
But the authors miss several important references in the first round review.
This should be added before the paper is published.

Reviewer: 1

Comments to the Author(s)

The authors satisfactorily implemented my suggestions.

Author's Response to Decision Letter for (RSOS-190671.R1)

See Appendix B.

Decision letter (RSOS-190671.R2)

03-Oct-2019

Dear Mr Dorraki,

I am pleased to inform you that your manuscript entitled "Parameter identification using moment of velocity (MoV)" is now accepted for publication in Royal Society Open Science.

Kind regards,
Anita Kristiansen
Royal Society Open Science Editorial Office
Royal Society Open Science
openscience@royalsociety.org

on behalf of R. Kerry Rowe (Subject Editor)
openscience@royalsociety.org

Appendix A

Manuscript ID: RSOS-190671

Title: Parameter identification using moment of velocity (MoV)

Author: Dorraki *et al*

To: R. Kerry Rowe, Editor

Dear R. Kerry,

Re: Reply to reviewers on MS #122615

Many thanks for reviewing our paper RSOS-190671. We have pleasure in attaching an updated manuscript and our point-by-point response to the comments is given below.

Best regards,

Mohsen Dorraki

Reviewer#1: The authors develop a new parameter identification method for system identification. The goal of the authors is laudable. However, I am not convinced by the example shown. The investigators should show how this works on real data. They need to provide concrete examples. They also need to use real noise. The simulation does not show how the system will work on actual data.

Author reply: We agree, and we thank the reviewer for this comment.

Author action: We have added a new subsection exploring the performance of MoV parameter identification on real world problems. This subsection examines the MoV on experimental responses from free vibration and industrial production, as follows:

(b) Real world problems

To assess the proposed parameter identification method for real world problems, two time series related to a free vibration test and an industrial production response after the 1990 shock are employed.

A free vibration test of the submerged pipeline is established by giving an initial displacement in the middle of the pipeline, and the response is recorded over time [26]. The damping coefficient $\alpha = 0.4824$ and the damped oscillation frequency $\omega_d = 20.1062$ are measured from the experiment. The time series is shown in Fig.3(a), and the corresponding MoV and fitted line are also illustrated in Fig.3(b and c). The estimated damping coefficient for this time series is obtained $\alpha_{est} = 0.4725$ and the estimated oscillation frequency is obtained $\omega_{est} = 20.3184$."

Figure 3: (a) The response corresponding to a free vibration test with damping coefficient $\alpha = 0.4824$ and oscillation frequency of $\omega_d = 20.1062$. (b) The moment of velocity of the free vibration response. (c) Taking logarithm from MoV and fitting a line on outcome signal. (d) An industrial production time series with damping coefficient $\alpha = 0.0077$ and oscillation frequency of $\omega_d = 0.1387$. (e) The moment of velocity of the industrial production time series. (f) Taking logarithm of MoV and fitting a line on the outcome signal.

In the same way, another dynamic response related to the industrial production response after the 1990 shock is used for assessing the MoV parameter identification approach. The oscillatory behavior of industrial production in Fig. 3(d). The damping coefficient $\alpha = 0.0077$ and the damped oscillation frequency $\omega_d = 0.1387$ are obtained [27]. Applying the MoV method to the time series provides the estimated damping coefficient $\alpha_{est} = 0.0061$ and the estimated oscillation frequency $\omega_{est} = 0.1770$. The MoV and the corresponding fitted line are shown in Fig. 3(e and f).

We also evaluate both MoV and the approach of Agneni et al. [14] as a well-known traditional parameter identification method and modified version of [12,13]. The dynamic properties of the mechanical and economic systems are listed in Table 1, which were obtained through experimental testing and parameter estimation. The performance of MoV versus the method of Agneni *et al.* is evaluated in the Table 1 that shows MoV is more reliable than Agneni's approach particularly for the industrial production case.

Table 1: The experimental and estimated damping parameters using our proposed MoV approach and a traditional method [14] for free vibration test [26] and industrial production [27] problems

Parameters	Free vibration			Industrial production		
	Experimental [26]	MoV	Traditional [14]	Experimental [27]	MoV	Traditional [14]
α	0.4824	0.4725 err= 2.05%	0.4618 err= 2.31%	0.0077	0.0061 err= 18.0328%	0.0057 err= 26.3158%
ω_d	20.1062	20.3184 err= 1.04%	19.8520 err= 1.28%	0.1387	0.1770 err= 21.6384%	0.1038 err= 33.6224%

Reviewer#1: There are also some grammatical errors which need remedy for example - even under the noisy conditions, even under noisy conditions. Matlab should be MATLAB.

Author reply: We agree.

Author action: We have corrected the grammatical errors containing those you mentioned here.

Reviewer#1: 'A limitation of the method is that errors become large in the 0 dB to 30 dB SNR range.' Authors need to show what happens and why and how to avoid it.

Author reply: As expected, it can be seen that a decrease in SNR results in increased error. Although MoV cannot cancel the impact of noise completely, it is more robust to noise in comparison of other methods. It is mentioned in literature that previous methods use instantaneous frequency which is more vulnerable than MoV in noisy conditions. A comprehensive noise analysis is carried out in [14] for both instantaneous frequency and MoV. The analysis shows how MoV outperforms instantaneous frequency in noisy conditions.

Author action: We have now altered the sentence "The noise may exert a considerable impact on the outcome of the parameter identification approach." To now read "The noise may exert considerable impact on the outcome of the parameter identification approach. A comprehensive noise analysis is presented in [14] for MoV and other Hilbert-based tools such as instantaneous frequency that explains the relative robustness of MoV in noisy conditions.... "

Reviewer#2: 1. The paper discusses the parameter identification problems. In fact, there exist many identification methods in the literature. Typical methods include the iterative identification methods [i1-i5] and the recursive identification methods [r1-r5]. These important works should be surveyed and mentioned in the paper, they play a very important role in system identification and parameter estimation. 2. Specifically, some classic transfer function parameter identification methods are very related to the authors' current submission, they are very useful for system analysis and control [t1-t5].

Author reply: We agree.

Author action: We have now added the relevant references.

Reviewer#2: 3. The paper is organized well and the paper contains some materials worth of publication.

Author reply: We thank the reviewer for this kind comment. We agree that this paper will be of wide interest.

Author action: No action.

Reviewer#2: 4. Some references are too old and may be removed. The references should use the RSOS style. 5. The paper can be accepted after revision according to the above comments.

Author reply: We agree.

Author action: The references are updated based RSOS format now.

Reviewer#3: The manuscript introduces an interesting method for the characterisation of a second order impulse response, using the Hilbert transform and the Moment of Velocity. The method appears to be novel, it is certainly new to me. The description of the techniques employed is clear and concise, and easy to follow. The results are good and the method certainly shows promise. The only thing that I suggest that the authors look to include is a direct comparison of their technique against a standard text book method.

Author reply: We agree. We evaluated both MoV and the approach of Agneni et al. [14] as a well-known traditional parameter identification method using real world responses.

Author action: We have now added a paragraph at the end of Section 5: "We also evaluate both MoV and the approach of Agneni et al. [14] as a well-known traditional parameter identification method and modified version of [12,13]. The dynamic properties of the mechanical and economic systems are listed in Table 1, which were obtained through experimental testing and parameter estimation. The performance of MoV versus the method of Agneni *et al.* is evaluated in the Table 1 that shows MoV is more reliable than Agneni's approach particularly for the industrial production case."

Table 1: The experimental and estimated damping parameters using our proposed MoV approach and a traditional method [14] for free vibration test [26] and industrial production [27] problems

Parameters	Free vibration			Industrial production		
	Experimental [26]	MoV	Traditional [14]	Experimental [27]	MoV	Traditional [14]
α	0.4824	0.4725 err= 2.05%	0.4618 err= 2.31%	0.0077	0.0061 err= 18.0328%	0.0057 err= 26.3158%
ω_d	20.1062	20.3184 err= 1.04%	19.8520 err= 1.28%	0.1387	0.1770 err= 21.6384%	0.1038 err= 33.6224%

Reviewer#3: In addition, I think that the paper would benefit from a comment on the robustness of the method if is applied to situations where the the impulse response is not strictly second order. Other than these two points, I would recommend publication.

Author reply: The proposed parameter identification method and other Hilbert-based approaches mentioned in literature are based on the assumption that the original system is a second order system, with an impulse response $y(t) = Ae^{-\alpha t}\cos(\omega_d t) + Be^{-\alpha t}\sin(\omega_d t)$, damping coefficient α , and oscillation frequency ω_d . For higher order systems, the impulse response contains multiple damping coefficients and multiple damping frequencies (depending on pole locations). Thus, these second order parameter identification methods do not work for the higher order systems. Please see the following Figures:

Fig. i: Pole-Zero map for a second order system $tf= 0.1s/(s^2+0.1s+1)$.

Fig. i shows a Pole-Zero map for the second order system we mentioned in the manuscript. It can be seen that the corresponding impulse response is $y(t)= 0.1 e^{-0.05t} \cos (t)$.

Fig. ii: Pole-Zero map for a fourth order system

Now we add a pair of poles to the last figure and make a fourth order system. Here, Fig. ii shows a Pole-Zero map for the fourth order system.

The impulse response for this system is $y(t) = c_1 e^{-0.05t} \cos(t) + c_2 e^{-0.025t} \cos(0.3t)$. It can be seen that the corresponding impulse response contain two different frequency components (1 and 0.3) and two different damping coefficients (0.05 and 0.025).

Author action: No action.

Appendix B

Manuscript ID: RSOS-190671.R1

Title: Parameter identification using moment of velocity (MoV)

Author: Dorraki *et al*

To: R. Kerry Rowe, Editor

Dear R. Kerry,

Re: Reply to reviewers on MS #122615

Many thanks for reviewing and accepting our paper RSOS-190671.R1. We have pleasure in submitting an updated manuscript.

Best regards,

Mohsen Dorraki